# Effects of Prolonged Serum Calcium Suppression during Extracorporeal Cardiopulmonary Resuscitation in Pigs

**DOI:** 10.3390/biomedicines11102612

**Published:** 2023-09-23

**Authors:** Jan-Steffen Pooth, Yechi Liu, Ralf Petzold, Christian Scherer, Leo Benning, Maximilian Kreibich, Martin Czerny, Friedhelm Beyersdorf, Christoph Benk, Georg Trummer, Sam Joé Brixius

**Affiliations:** 1Department of Emergency Medicine, Faculty of Medicine, University Medical Centre Freiburg, University of Freiburg, 79106 Freiburg, Germany; jan-steffen.pooth@uniklinik-freiburg.de (J.-S.P.); ralf.petzold@gmx.de (R.P.); leo.benning@uniklinik-freiburg.de (L.B.); 2Department of Cardiovascular Surgery, Faculty of Medicine, University Medical Centre Freiburg, University of Freiburg, 79106 Freiburg, Germany; yechi.liu@rkk-klinikum.de (Y.L.); christian.scherer@uniklinik-freiburg.de (C.S.); maximilian.kreibich@uniklinik-freiburg.de (M.K.); martin.czerny@uniklinik-freiburg.de (M.C.); friedhelm.beyersdorf@uniklinik-freiburg.de (F.B.); christoph.benk@uniklinik-freiburg.de (C.B.); georg.trummer@uniklinik-freiburg.de (G.T.)

**Keywords:** controlled reperfusion, CARL, animal model, cardiac arrest, extracorporeal resuscitation, ischemia–reperfusion injury, apoptosis, calcium

## Abstract

Controlled reperfusion by monitoring the blood pressure, blood flow, and specific blood parameters during extracorporeal reperfusion after cardiac arrest has the potential to limit ischemia–reperfusion injury. The intracellular calcium overload as part of the ischemia–reperfusion injury provides the possibility for the injury to be counteracted by the early suppression of serum calcium with the aim of improving survival and the neurological outcome. We investigated the effects of prolonged serum calcium suppression via sodium citrate during extracorporeal resuscitation using the CARL protocol (CARL—controlled automated reperfusion of the whole body) compared to a single-dose approach in a porcine model after prolonged cardiac arrest. A control group (N = 10) was resuscitated after a 20 min cardiac arrest, initially lowering the intravascular calcium with the help of a single dose of sodium citrate as part of the priming solution. Animals in the intervention group (N = 13) received additional sodium citrate for the first 15 min of reperfusion. In the control group, 9/10 (90.0%) animals survived until day 7 and 7/13 (53.8%) survived in the intervention group (*p* = 0.09). A favorable neurological outcome on day 7 after the cardiac arrest was observed in all the surviving animals using a species-specific neurological deficit score. The coronary perfusion pressure was significantly lower with a tendency towards more cardiac arrhythmias in the intervention group. In conclusion, a prolonged reduction in serum calcium levels over the first 15 min of reperfusion after prolonged cardiac arrest tended to be unfavorable regarding survival and hemodynamic variables compared to a single-dose approach in this animal model.

## 1. Introduction

Cardiac arrest is associated with a high mortality and remains one of the major medical challenges of our time, despite the increased emergence of first-responder alert systems, optimized rescue chains, extensive training programs, and the increased use of extracorporeal cardiopulmonary resuscitation (ECPR) systems. The survival to hospital discharge for in-hospital cardiac arrest (IHCA) is about 20% worldwide, while out-of-hospital cardiac arrest (OHCA) is survived by only 8–10%, depending on the region [1,2,3]. In addition, a significant number of survivors after resuscitation suffer from a variety of neurological impairments, including the permanent loss of brain function [4,5], with serious socioeconomic consequences. Recent studies have linked the unsatisfactory outcomes after cardiopulmonary resuscitation to ischemia–reperfusion injury (IRI), possibly due to the inadequate and/or uncontrolled restoration of circulation [6,7].

To be able to treat the effects of ischemia–reperfusion injury individually and to limit the reperfusion injury, the CARL (controlled automated reperfusion of the whole body) protocol was developed in large animal/porcine studies, which, in contrast to previously available methods, allows the monitoring and adjustment of essential physical and metabolic parameters during extracorporeal reperfusion [8,9]. The CARL protocol is essentially based on the implementation of extracorporeal circulation (ECC) with a high pulsatile blood flow, a high blood pressure, rapidly induced hypothermia, and the use of an optimized priming solution. The blood oxygen level can be individually adjusted to reduce the formation of radical oxygen species, and the ischemia-induced acidosis is gradually resolved by controlling carbon dioxide. In addition, permanent monitoring of the arterial pressure and a blood gas analysis are performed, in which all therapy-relevant venous and arterial parameters are closely recorded. The CARL protocol was developed using established principles of cardiac surgery and perfusion [9]. In a porcine animal model, the consistent application of the CARL protocol increased the survival rate after a 20 min normothermic cardiac arrest to more than 90%, and more than 90% of the surviving animals showed no or no significant neurological impairments after resuscitation [8,10].

The CARL protocol includes an initial lowering of the serum calcium levels by using sodium citrate in the reperfusate. The rationale for this is based on the observation that calcium in the ischemically damaged organism triggers several apoptotic and inflammatory signal transduction pathways that cause programmed cell death and, ultimately, the death of the entire organism (Figure 1) [6,11,12,13]. In cardiac arrest with subsequent resuscitation, the cytosolic calcium levels already increase during the ischemic phase [12]. The lack of oxygen initially induces a rapid depletion of energy carriers, which can no longer be compensated for by glycolysis. The cell then switches to anaerobic lactate metabolism, which subsequently contributes to the disruption of ion exchange across the cell membrane in such a way that extracellular calcium is ultimately transferred into the cell via the Na^+^/Ca^2+^ exchanger (sodium–calcium exchanger, NCX) [14,15,16,17,18]. This passive, inverted action of the NCX is often referred to as the “reverse mode”; it leads to sodium efflux and, in turn, allows the influx of extracellular calcium [14,19]. In addition, ischemia inhibits intracellular calcium reuptake into the endoplasmic reticulum, while the calcium storages are depleted and calcium accumulates intracellularly [12,14,20]. This cascade is exacerbated at the onset of reperfusion because the extracellular H^+^ concentration decreases as a result of the reperfusion, thereby further activating ion transport across the cell membrane in favor of additional calcium inflow [14,15,16]. This is then commonly referred to as a calcium overload. A physiological equilibrium of ion concentrations can only be re-established with physiological cellular metabolism in the further course of resuscitation [21].

About 40% of the plasmatic calcium ions (Ca^2+^) are bound to albumin and 13% are bound to endogenous anions. The physiologically active form (47%) is present as free, ionized calcium (iCa^2+^), which forms a chelate complex with sodium citrate or appropriate medication [22]. The metabolically active calcium content can be titrated accordingly with sodium citrate [23]. In the CARL protocol, the blood calcium level is initially lowered with the help of a priming solution, which contains sodium citrate.

The aim of this study is to clarify whether a prolonged reduction in the serum calcium level in the early phase of reperfusion can be realized safely without increasing the side effects of induced hypocalcemia, and whether this approach is associated with an additional benefit in terms of the overall and neurologically intact survival.

## 2. Materials and Methods

The presented study is a controlled, experimental animal study conducted in pigs (German Landrace, see Figure 2). The animals were, on average, 5 months old and weighed 53.3 ± 3.7 kg (minimum weight: 48.0 kg, maximum weight: 62.0 kg). The experiments were approved by the Regierungspräsidium Freiburg (protocol numbers G-10/90 and G-15/148) according to the German Animal Welfare Act and were performed according to the rules and regulations of the European Union (2010/63/EU).

### 2.1. Preoperative Management and Anesthesia

Our animal model has been described in detail in previous publications [8,10]. In short, the experimental animals were held in the animal facility of the University Medical Centre Freiburg one week before the experiment and were initially sedated with ketamine (20 mg/kg body weight (bw)) and midazolam (0.5 mg/kg bw) after 12 h of fasting. The lateral auricular vein was catheterized, and the animals were preoxygenated with an oxygen mask for 3 min. The animals then received an intravenous bolus of 1% propofol (2–4 mg/kg bw) until the loss of the swallowing reflex, after which they were endotracheally intubated and connected to intermittent positive pressure ventilation (Cato, Draeger, Lübeck, Germany). The oxygenation was adapted to aim for an arterial partial pressure of oxygen of 80–100 mmHg. Muscle relaxation was achieved with vecuronium (0.2 mg/kg bw), a gastric tube was inserted, and the animals were positioned supine on the operating table. The anesthesia was initially maintained with 1–2 vol% isoflurane and switched to 1% propofol intravenously (i.v.) at 10–15 mg/kg bw/h with the onset of reperfusion. For infection prophylaxis, 2 g of ceftriaxone was administered.

### 2.2. Blood Sampling and Induction of Cardiac Arrest

The internal jugular vein and the common carotid artery were cannulated for blood sampling. An additional cannula for continuous blood pressure measurements was inserted further distally into the common carotid artery (20 G Leader-Cath arterial catheter, Vygon, Ecouen, France). A Swan-Ganz catheter was inserted into the subclavian vein to measure the central venous pressure (CVP) (7 French, Arrow, PA, USA). The systolic, diastolic, and mean blood pressures and the CVP were measured continuously and documented every 4 s.

Cardiac arrest was induced electrically via a flicker electrode (FI-10m, Stöckert, Sorin-Group, Munich, Germany), which was inserted following an epigastric incision. Before the induction of cardiac arrest, the animals received heparin i.v. (300 I.U./kg bw) to prevent the clotting of the cannulas during the arrest phase.

### 2.3. Controlled Reperfusion (CARL Protocol) and Citrate Addition

During the 20 min cardiac arrest, the extracorporeal circulation system was implemented via the femoral artery and external jugular vein (arterial cannula 14–16 French, venous cannula 22–24 French; LivaNova Deutschland GmbH, Munich, Germany). The duration of 20 min was chosen in accordance with the current consensus statement for ECPR of the Extracorporeal Life Support Organization, which states that ECPR may be justified after 20 min of refractory arrest [24].

The CARL protocol consists of controlled reperfusion by controlling the blood flow and pressure parameters as well as monitoring the venous and arterial parameters via a blood gas analysis and an elaborate priming solution [9].

Immediately after the start of reperfusion, the animals received 40 mL of 7.45% potassium chloride for cardioversion. If necessary, this procedure was performed twice and, if unsuccessful, was followed by electrical cardioversion, as previously described by our group [25]. In the intervention group, additional sodium citrate was administered during the first 15 min of reperfusion to aim for an arterial ionized calcium level < 0.6 mmol/L.

The reperfusion flow was measured and recorded every 10 s. Blood was repeatedly drawn for blood gas analyses from the common carotid artery of all experimental animals up to one hour after the end of reperfusion (Cobas b 123, Roche Diagnostics Deutschland GmbH, Mannheim, Germany). The coronary perfusion pressure (CPP), defined as the difference between the diastolic arterial blood pressure and the central venous pressure, was calculated to assess the cardiac perfusion. Norepinephrine was administered to maintain a mean arterial blood pressure above 65 mmHg. The occurrence of cardiac arrhythmias was monitored and documented using an electrocardiogram (SC 9000XL, Siemens AG, Erlangen, Germany).

Extracorporeal perfusion was maintained for 60 min, after which the animals were weaned from the extracorporeal circulation. The primary survival was defined as successful weaning from ECC. After weaning, heparin was antagonized with an intravenous protamine administration (200 I.U./kg bw).

### 2.4. Postoperative Care and Clinical Assessment

Postoperatively, oxygenation was ensured by synchronized intermittent ventilation, and carprofen was administered i.v. (4 mg/kg bw) for pain relief. The animals were placed on their abdomen and extubated as soon as the swallowing reflex and spontaneous breathing appeared regularly. When the oxygen saturation and heart rate were stable for more than one hour, the pigs were moved to the animal facility and closely monitored for the following 7 days. The neurological outcome was measured daily by a veterinarian with a species-specific neurological deficit score (NDS) [26]. An NDS below 50 was considered a favorable neurological outcome. After the NDS assessment on day 7, the animals were euthanized by a potassium overdose under deep propofol narcosis.

### 2.5. Statistical Analysis

The statistical analysis and visualization were performed using SigmaPlot, version 13 (Systat Software GmbH, Erkrath, Germany) and R statistical software (version 4.2.1) [27]. To control for the assumption of normally distributed data, a Shapiro–Wilk test was performed. To test for differences between groups, Mann–Whitney U tests and Student’s *t*-tests were used as appropriate. Statistical analyses regarding blood pressure and coronary blood pressure were conducted in a mixed-effects model for each phase of the experiment using the R packages lme4 and lmerTest. For categorical data, Fisher’s exact test was applied. *p*-values < 0.05 were considered statistically significant. Given two groups with a sample size of N = 10 and N = 13, a difference in means could be detected with an effect size greater than 1.2, a power of 0.8, and a two-tailed significance value of 0.05.

## 3. Results

### 3.1. Ionized Arterial Calcium Levels

The arterial blood gas analysis revealed no differences in the baseline values for the ionized calcium levels between the two groups (intervention group: 1.41 ± 0.04 mmol/L vs. control group: 1.39 ± 0.04 mmol/L, *p* = 0.43). The total serum calcium levels also did not differ between the groups at baseline (intervention group: 2.52 ± 0.12 mmol/L vs. control group: 2.46 ± 0.14 mmol/L, *p* = 0.25). As shown in Figure 3, during reperfusion, the intervention group consistently showed significantly lower ionized calcium values compared to the control group.

Appropriately, the minimum calcium level was found to be lower in the intervention group than in the control group (0.43 ± 0.17 mmol/L vs. 0.58 ± 0.08 mmol/L, *p* < 0.05). Different ionized calcium levels could be detected for up to 2 h after the start of reperfusion (intervention group: 0.90 ± 0.18 mmol/L vs. control group: 1.15 ± 0.14 mmol/L, *p* < 0.001).

### 3.2. Hemodynamics and Norepinephrine Requirement

Continuous measurements of the arterial blood pressure during ECC revealed a significant difference within the first 30 min of reperfusion between the two groups for the systolic (*p* = 0.01), diastolic (*p* = 0.02), and mean arterial pressure (*p* = 0.03) (Figure 4A).

Looking at the mean values of the coronary perfusion pressure, a significant difference between the control and the intervention group within the first hour of reperfusion could be observed, with higher values for the control group (89.9 ± 25.2 mmHg vs. 103.1 ± 20.2 mmHg, *p* = 0.03) (Figure 4B). There was no significant difference in the norepinephrine requirements between the control and intervention groups (7.08 ± 3.45 μg/kg vs. 9.54 ± 4.93 μg/kg, *p* = 0.37) (Figure 4C).

Due to a technical defect, no data were available regarding the CPP and MAP of one animal in the control group and the noradrenaline requirement for one animal from each group.

### 3.3. Cardiac Arrhythmias and Bleeding

After the initial successful cardioversion after weaning from ECC, a trend towards an increased rate of arrhythmias was noticed in the intervention group, as shown in Table 1. Since the ECG used did not allow for the recording of the ECG leads, a more detailed differentiation of the observed arrhythmias was not performed. In two animals, the postoperative arrhythmias could only be terminated via the administration of intravenous calcium. One animal in the control group died of malignant arrhythmias within the first hours after weaning from ECC.

An increased bleeding propensity was not observed postoperatively in any of the animals. Appropriately, there were no group differences between the hemoglobin concentrations after weaning from ECC (intervention group: 8.6 ± 0.9 mg/dL vs. control group: 9.3 ± 0.9 mg/dL, *p* = 0.37), at the end of the experiment (8.6 ± 0.8 mg/dL vs. 8.6 ± 0.7 mg/dL, *p* = 1), or at baseline (9.5 ± 0.7 mg/dL vs. 9.6 ± 0.8 mg/dL, *p* = 1) (Figure 4D).

### 3.4. Survival and Neurological Deficit

The overall survival and neurological outcome are shown in Figure 5. All animals survived the 20 min cardiac arrest after being treated according to the CARL protocol (Figure 5A). However, several hours after weaning from ECC, two animals in the intervention group died of refractory pulmonary edema and one animal from the control group died of malignant cardiac arrhythmias. Subsequently, four animals in the intervention group were euthanized prematurely because the predefined termination criteria were met.

Thus, a total of 9 out of 10 animals from the control group (90.0%) and 7 out of 13 animals from the intervention group (53.8%) survived until day 7. These results were not statistically significant (*p* = 0.08).

The neurological outcome measured by the NDS showed a good outcome (defined as an NDS < 50) in all animals that survived until day 7 of the experiment, with no difference between both groups (*p* = 0.87) (Figure 5B).

## 4. Discussion

As mentioned above, an intracellular calcium overload plays an important role in the development of ischemia–reperfusion injury after global ischemia, which finally contributes to a poor outcome after cardiopulmonary resuscitation. This phenomenon may be counteracted by lowering the serum calcium levels in an early phase of extracorporeal resuscitation.

The presented work confirms the results of previous studies on the efficacy of a single dose of sodium citrate for initial calcium lowering in vivo [8,10]. The continued application of sodium citrate over 15 min, in addition to the initial priming dose, led to a greater reduction in the arterial ionized calcium levels with a constant low level of ionized calcium over the time of the application (Figure 3). At minute 10 after reperfusion from cardiac arrest, the iCa^2+^ concentrations in the two groups varied by about 0.3 mmol/L, and this difference was apparent until the end of reperfusion (Figure 3). In both groups, the calcium level spontaneously rose again due to redistribution processes, as already seen in previous studies [8,10]. The prolonged lowering of ionized calcium levels with sodium citrate thus had a higher effect compared to a single-dose application.

In this context, it is important to note that the first minutes of reperfusion are crucial for the containment of the reperfusion injury [6,28]. After this phase, a point of no return seems to be reached, after which the damage may outweigh the benefit of prolonged serum calcium lowering. The organism relies on a physiologically balanced calcium level to reactivate important repair processes, signal transduction cascades, and energy and cellular metabolism. For example, the vascular wall muscles require sufficient calcium to maintain the vascular tonus, and the heart needs calcium for rhythm stabilization [29]. Furthermore, ionized calcium plays a central role in the coagulation cascade. Known as factor IV, calcium is involved in many steps of the coagulation cascade and is thus an important part of hemostasis [30]. The timeslot for a potential positive effect of a reduction thus seems to be limited to the initial reperfusion, with a whole series of potential side effects through a sustained lowering. In this context, studies of ischemic stroke show that a low calcium concentration at hospital discharge is a negative prognostic factor for the infarct area, neurological recovery, and survival [31,32].

Regarding survival, this study confirmed the results of previous studies on the efficiency of the CARL protocol in animal experiments [8,10]. Shortly after the initiation of controlled reperfusion, 90% of the animals in the control group regained sinus rhythm and survived weaning from extracorporeal circulation. In the control group, one animal died of malignant arrhythmias within the first hours. Nonetheless, the survival rate one week after a 20 min normothermic cardiac arrest followed by treatment according to the CARL protocol was still 90% in the control group, with all the animals showing a good neurological outcome (NDS < 50, Figure 5B).

In the intervention group with prolonged calcium suppression, only 53.8% of all the animals survived (Figure 5A). Although no statistical significance was found, the lower survival rate possibly provides a first indication that an additional suppression of the blood calcium level during the first 15 min of extracorporeal reperfusion may not provide an additional benefit, but rather impairs the prognosis.

This hypothesis is strengthened by a non-superior neurological outcome shown in the presented data. Despite prolonged serum calcium depletion, no accelerated neurological recovery was seen in the intervention group (Figure 5B).

Overall, the survival rate and neurological outcome in this animal model far exceeded those in the human setting, where even patients who suffer cardiac arrest in-hospital survive to hospital discharge in only about 20% of the cases worldwide [33,34]. However, the first human applications of the CARL protocol suggest that the prognosis after cardiac arrest may be improved, particularly for the treatment of OHCA using new technologies [9,35].

During the reperfusion phase, both groups achieved a high MAP of above 80 mmHg as provided in the CARL protocol (Figure 4A). To achieve these reperfusion pressures, comparable doses of norepinephrine were used in both groups (Figure 4C). An adequate perfusion pressure is important to ensure perfusion, especially of microcapillaries in the brain, and to counteract the cerebral no-reflow phenomenon [6,36,37]. Clinical studies have shown that a high MAP and CPP are, together with a high systolic blood pressure, associated with a lower mortality and better neurological outcomes after cardiac arrest [38,39]. Regarding blood pressure, significant differences between the two groups, especially in the beginning of reperfusion and during calcium lowering, were observed in this study (Figure 4A). A calcium-dependent impact on blood pressure and myocardial inotropy is well known [40]. Calcium lowering significantly affects blood pressure, since blood pressure is predominantly regulated by peripheral vascular resistance. To maintain this, the vascular wall muscles require a sufficient level of calcium.

Possibly, the observed calcium-related reduction in blood pressure may have had a negative effect on the neurological outcome in the intervention group, thus directly counteracting the desired beneficial effect by limiting the calcium overload.

The animals in the intervention group showed a significantly lower mean coronary perfusion pressure than those in the control group (Figure 4B). Early and adequate coronary reperfusion is important for functional recovery and for preventing myocardial damage after cardiac arrest [41,42,43]. Accordingly, the presented study found an increase in arrhythmias in the intervention group, which, on the one hand, could be interpreted as a direct effect of the proarrhythmogenic effects of hypocalcemia, and, on the other hand, as a consequence of the lower CPP, as already assumed in preliminary studies. The determination of circulating calcium receptors in serum has already been clinically linked to the treatment success of arrhythmias and might represent an important parameter for future research to further elucidate this observation [44].

Despite iatrogenic calcium lowering, no increased bleeding complications were observed in this study and the hemoglobin concentrations were found to be comparable between both groups. However, it must be noted at this point that, due to the therapeutic anticoagulation and surgical vascular access, the study design was probably unsuitable to evaluate and detect an increased bleeding propensity or bleeding complications.

### Limitations

The study limitations include the use of juvenile pigs and the administration of volatile anesthetics, which may have contributed to improved survival in both experimental groups. Since pigs above a body weight of 60 kg were too heavy for one person to handle in the stable during the intensive 24/7 post-resuscitation care, we were forced to rely on juvenile animals for our experiments.

## 5. Conclusions

The results of this study suggest that an initial lowering of the ionized calcium concentration is feasible and may contribute to the limitation of ischemia–reperfusion injury in controlled reperfusion by limiting the calcium overload after prolonged cardiac arrest. Nevertheless, it represents a major intervention in a life-threatening and unstable situation and should, therefore, only be applied under close monitoring and for a minimal period of time. The prevalence of arrhythmias, a reduced vascular resistance with an elevated catecholamine demand, and an increased bleeding propensity should be anticipated and treated accordingly. Our study showed that a prolonged reduction in calcium levels over the first 15 min of reperfusion after prolonged cardiac arrest tended to be unfavorable regarding the survival and hemodynamic variables compared to a single-dose approach.

## Figures and Tables

**Figure 1 biomedicines-11-02612-f001:**
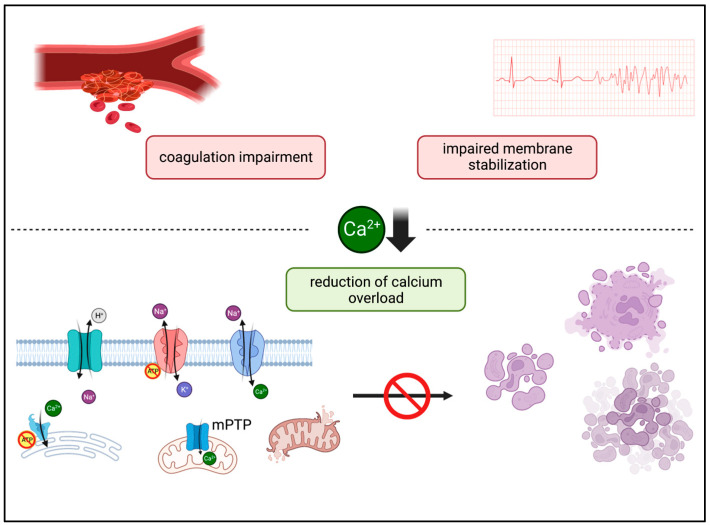
Potential desirable and adverse effects of calcium lowering in reperfusion after ischemia. mPTP: mitochondrial permeability transition pore.

**Figure 2 biomedicines-11-02612-f002:**
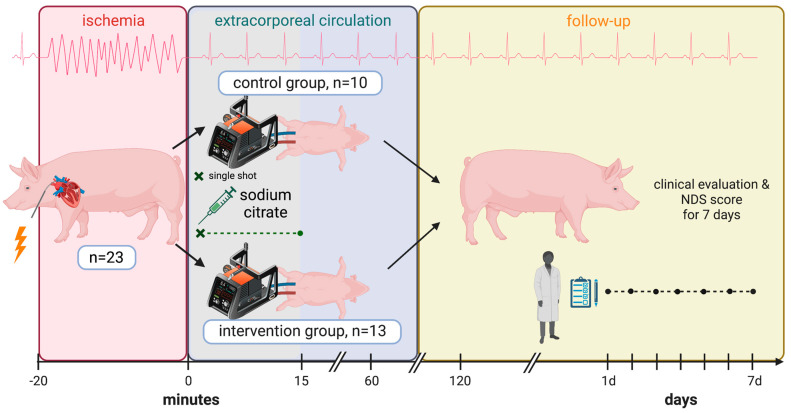
Schematic presentation of the experimental protocol. NDS: neurological deficit score.

**Figure 3 biomedicines-11-02612-f003:**
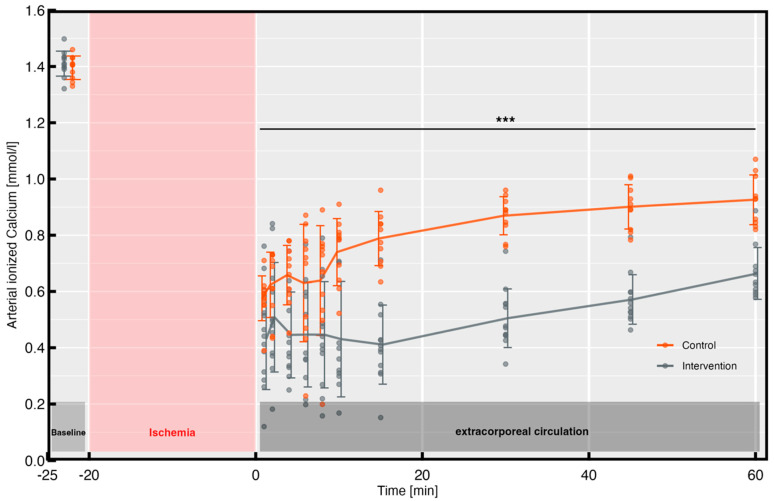
Arterial ionized calcium during the first hour of reperfusion. *** *p* < 0.001.

**Figure 4 biomedicines-11-02612-f004:**
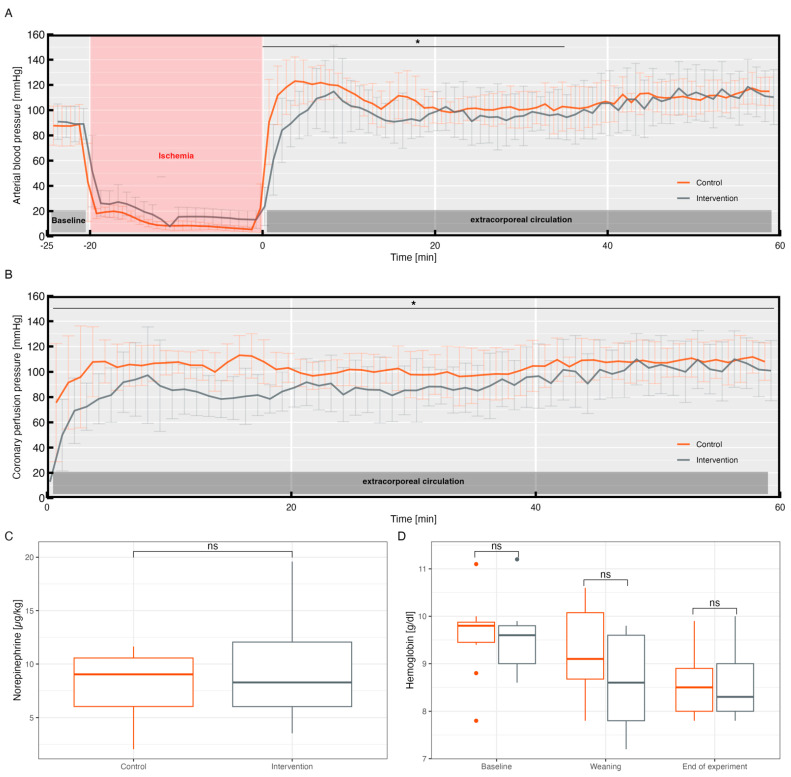
(**A**): Arterial blood pressure during baseline and first hour of reperfusion. (**B**): Coronary perfusion pressure during the first hour of reperfusion. (**C**): Total amount of norepinephrine used during the experiment. (**D**): Hemoglobin levels at baseline, weaning from extracorporeal circulation, and end of experiment. * *p* < 0.05. ns = not significant.

**Figure 5 biomedicines-11-02612-f005:**
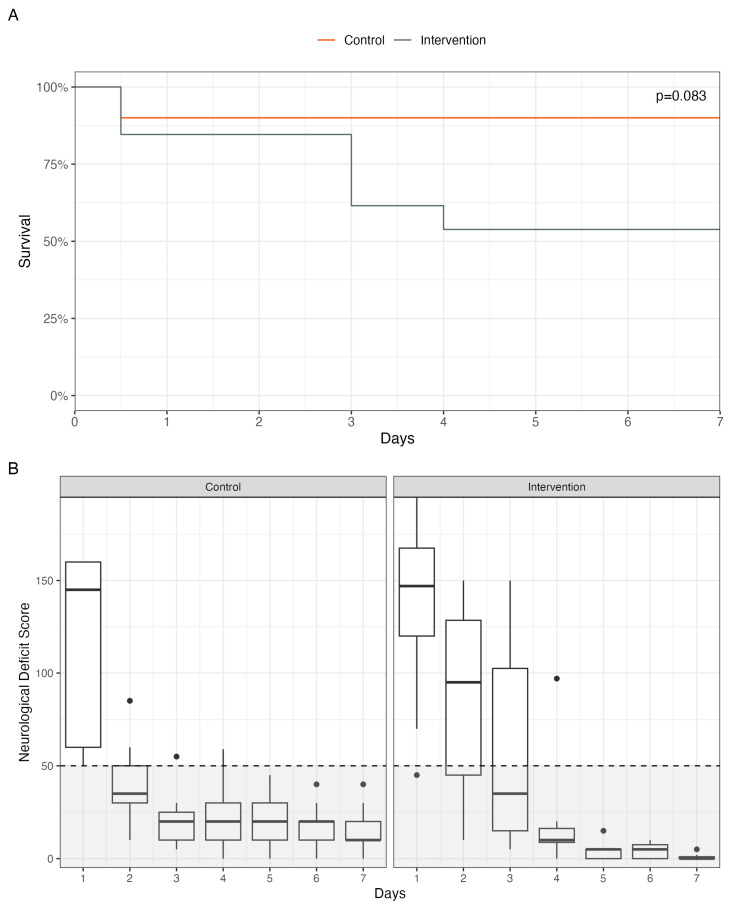
(**A**): Kaplan–Meier curves for each group. (**B**): Neurological deficit scores (NDS) of the two groups during the follow-up period. Dotted lines mark an NDS of 50. An NDS below 50 was predefined as a favorable neurological outcome (gray area).

**Table 1 biomedicines-11-02612-t001:** Prevalence of arrhythmias.

Group	Arrhythmias During ECC		Postoperative Arrhythmias	
	N	*p*-Value	N	*p*-Value
**Control**, N = 10	4/10	0.10	2/10	0.66
**Intervention**, N = 13	10/13		4/13	

## Data Availability

All the analyzed datasets are available in anonymous form from the corresponding author upon reasonable request.

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
