# Peer review of "Effects of Prolonged Serum Calcium Suppression during Extracorporeal Cardiopulmonary Resuscitation in Pigs"

_biomedicines, 2023, doi:10.3390/biomedicines11102612_

Round 1

Reviewer 1 Report

In this experimental study, Dr. Pooth and colleagues investigated the effect of prolonged calcium suppression during ECPR in porcine models. Overall, this is an interesting study with potential clinical applications. Nonetheless, I have a few questions:

1) In the title, the authors need to mention which calcium they suppressed? serum or intracellular or maybe extracellular?

2) "The intracellular calcium overload as part of the ischemia-reperfusion injury provides the possibility to be counteracted by an early calcium suppression with the aim of improving survival and neurological outcome.", again which calcium? Also, in the text, the authors need to explain how ischemia reperfusion can induce intracellular calcium overload? Through which mechanisms? sarcoplasmic reticulum calcium release or intracellular influx? This review might be helpful (PMID: 32188566)

3) I am wondering if the authors checked the serum (both total and ionized) calcium before cardiac arrest and before ECPR? How many of the animals were initially hypocalcemic? or maybe some of them were hypercalcemic?

4) In Figure 3 for example, the data began at 20 minutes of ECPR? What about the 0-20 minutes timeframe? Please also show the Ca concentration pre cardiac arrest, baseline calcium (before sodium citrate application) and 0-20 minutes after application of either one time or prolonged sodium citrate application. 

5) Why did the authors perform ECPR after 20 minutes of cardiac arrest? Please explain, as the timing is very crucial for survival. The longer the delay, the lower the survival. 

6) Is the application of a single dose sodium citrate based on an established concensus or guideline or a common practice? I think the authors need to add an additional control group with no sodium citrate.

7) "Favorable neurological outcome at day 7 after cardiac arrest was observed in all surviving animals." using? please mention NDS here. What are the main findings? Please also add in the abstract. 

8) "The coronary perfusion pressure showed to be significantly lower with a tendency to more cardiac arrhythmias in the intervention group." which cardiac arrhythmias? tachy, brady, atrial, ventricular?

9) "The cell then switches to anaerobic lactate metabolism, which subsequently disrupts ion exchange across the cell membranes in such a way that calcium is ultimately transferred into the cell via the Na+/Ca2+ exchanger (sodium-calcium exchanger, NCX)" Does this mean that during anaerobic respiration, NCX works in a reverse mode? Please explain and provide some supporting references. The authors can also read this publication (PMID: 33486968)

10) "In addition, ischemia inhibits Ca2+ reuptake into the endoplasmic reticulum (ER), while calcium release from the ER is accelerated" How could it be accelerated? via which mechanisms? Also, if the uptake via SERCA is diminished then the calcium storage will be depleted, thereafter the Ca release via ryanodine will be decreased.

11) "The aim of this study is to clarify whether a further reduction of the calcium level in the early phase of reperfusion can be realized safely without increasing the side effects of induced hypocalcemia" How big is this extra reduction? Please quantify.

12) In section 3.3, please specify the type of arrhythmia observed. If possible, please also include the ECG tracing.

No comment

Author Response

In this experimental study, Dr. Pooth and colleagues investigated the effect of prolonged calcium suppression during ECPR in porcine models. Overall, this is an interesting study with potential clinical applications. Nonetheless, I have a few questions:

Thank you for reviewing our manuscript.

1) In the title, the authors need to mention which calcium they suppressed? serum or intracellular or maybe extracellular?

The title was amended accordingly.

2) "The intracellular calcium overload as part of the ischemia-reperfusion injury provides the possibility to be counteracted by an early calcium suppression with the aim of improving survival and neurological outcome.", again which calcium? Also, in the text, the authors need to explain how ischemia reperfusion can induce intracellular calcium overload? Through which mechanisms? sarcoplasmic reticulum calcium release or intracellular influx? This review might be helpful (PMID: 32188566)

Thank you for giving us the opportunity to include more information regarding calcium overload in ischemia and reperfusion. We have revised the abstract on lines 18 and 20 accordingly. As described in the cited references (PMID: 21856909 [ref #16] and 20167368 [ref #14]) intracellular calcium overload is caused by NCX mitigated intracellular calcium influx as well as calcium release from intracellular sources. We provide more detailed information on lines 66-83 and have amended this part accordingly.

3) I am wondering if the authors checked the serum (both total and ionized) calcium before cardiac arrest and before ECPR? How many of the animals were initially hypocalcemic? or maybe some of them were hypercalcemic?

Thank you for this comment. The ionized calcium at baseline is reported on lines 186-188 and showed no differences. Since only ionized calcium is biologically active, we initially chose to only report the ionized calcium. We have now also included total serum calcium levels at baseline. No difference was detected regarding total calcium at baseline. This information and the corresponding total serum calcium levels were added on lines 188-190.

4) In Figure 3 for example, the data began at 20 minutes of ECPR? What about the 0-20 minutes timeframe? Please also show the Ca concentration pre cardiac arrest, baseline calcium (before sodium citrate application) and 0-20 minutes after application of either one time or prolonged sodium citrate application. 

Thank you for making us aware of this inconsistency regarding the displayed timelines. In Figure 2 “start of extracorporeal circulation” was displayed as minute 0. In Figures 3 and 4 “start of ischemia” was chosen as minute 0 and therefore “start of extracorporeal circulation” was displayed as minute 20. We have now revised the timelines in Figure 3 and Figure 4 accordingly. All Figures now display “start of extracorporeal circulation” as minute 0. Figure 3 now also includes baseline measurements.

5) Why did the authors perform ECPR after 20 minutes of cardiac arrest? Please explain, as the timing is very crucial for survival. The longer the delay, the lower the survival.

Thank you for this question. The reviewer is correct that longer periods of ischemia result in lower survival. As stated in the current AHA and ERC guidelines for resuscitation, ECPR should be considered, when conventional resuscitation is failing. The current ELSO consensus states that ECPR may be justified after 20 minutes of refractory arrest (PMID: 33627592 [ref #24]). We have included this information now on lines 138-141.

6) Is the application of a single dose sodium citrate based on an established concensus or guideline or a common practice? I think the authors need to add an additional control group with no sodium citrate.

We thank the reviewer for this valuable suggestion. Since the application of a single dose of sodium citrate is already clinical practice within the CARL protocol, which is used in many centers across Europe, the aim of this study was to evaluate a prolonged reduction of serum calcium levels. Therefore, we did not add an additional control group without sodium citrate. Clinical results of the CARL protocol have been evaluated in a European multi-center study (https://drks.de/search/en/trial/DRKS00018967) and its preliminary results are currently under review in a different journal.

7) "Favorable neurological outcome at day 7 after cardiac arrest was observed in all surviving animals." using? please mention NDS here. What are the main findings? Please also add in the abstract.

Thank you for giving us the opportunity to revise this part of the manuscript. We have now included the neurological deficit score on lines 27-28 as recommended and added a conclusion in the abstract on lines 29-32.

8) "The coronary perfusion pressure showed to be significantly lower with a tendency to more cardiac arrhythmias in the intervention group." which cardiac arrhythmias? tachy, brady, atrial, ventricular?

Thank you for this comment. ECG monitoring was used in all animals but unfortunately the ECG was not recorded. The anesthesiologist only marked the occurrence and timing of any arrhythmia in his documentation. Therefore, we are unfortunately unable to provide the requested information. We now acknowledge the inability to further classify the cardiac arrhythmias on lines 220-221.

9) "The cell then switches to anaerobic lactate metabolism, which subsequently disrupts ion exchange across the cell membranes in such a way that calcium is ultimately transferred into the cell via the Na+/Ca2+ exchanger (sodium-calcium exchanger, NCX)" Does this mean that during anaerobic respiration, NCX works in a reverse mode? Please explain and provide some supporting references. The authors can also read this publication (PMID: 33486968)

We thank the reviewer for this comment. During reperfusion NCX, which “can move calcium ions in either direction depending on the resting membrane potential and the transmembrane gradients of sodium and calcium” (PMIDs: 27526656 [ref #20] and 1996686), does work in a reverse mode (PMID: 11470463). Besides the already cited references we now also refer to this publication (PMID: 27526656 [ref #20]). Anaerobic respiration is of course not the only reason for this reverse mode of action. We have clarified this sentence accordingly and added the before mentioned publication as a reference. Changes were made on lines 72-77.

10) "In addition, ischemia inhibits Ca2+ reuptake into the endoplasmic reticulum (ER), while calcium release from the ER is accelerated" How could it be accelerated? via which mechanisms? Also, if the uptake via SERCA is diminished then the calcium storage will be depleted, thereafter the Ca release via ryanodine will be decreased.

Thank you for giving us the opportunity to clarify this part of the manuscript. We meant to describe the redistribution of intracellular calcium due to the lack of energy carriers. We have revised this sentence on lines 77-79 accordingly.

11) "The aim of this study is to clarify whether a further reduction of the calcium level in the early phase of reperfusion can be realized safely without increasing the side effects of induced hypocalcemia" How big is this extra reduction? Please quantify.

Thank you for this comment. The use of “further” in this context was misleading. We did not intend a further reduction but rather a prolonged reduction of serum calcium as shown in Figure 3. The above-mentioned sentence was revised accordingly on line 95.

12) In section 3.3, please specify the type of arrhythmia observed. If possible, please also include the ECG tracing.

Thank you for this suggestion. As mentioned above we are unfortunately unable to provide the requested information. We now acknowledge the inability to further classify the cardiac arrhythmias on lines 220-221.

Reviewer 2 Report

This is a well performed experiment. The intervention with sodium citrate does not result in an improved outcome. It is worthwhile to publish these 'negative' results as a cautionary note for future treatment of cardiac arrest, which still has a poor outcome. 

line 201: MAD or MAP? This should be checked

Author Response

Thank you very much for this review.

line 201: MAD or MAP? This should be checked

We have corrected this line. It now reads MAP.

Reviewer 3 Report

I read with great pleasure the article by Jan-Steffen Pooth and colleagues titled "Effects of prolonged serum calcium suppression during extracorporeal cardiopulmonary resuscitation in pigs". As a person who conducts similar research in the field of emergency medicine on an animal model, I must congratulate the authors on a very good work and a wonderful presentation of it in the manuscript and figures included in the work. I had the advantage of also seeing the previous version of the manuscript, and basically everything I had comments on was included in the new version. Personally, I don't see anything that I could comment on at the moment - the introduction is complete and covers the topics discussed by the authors very broadly. The methods and results are described extensively and very thoroughly, for which the authors should be congratulated once again. The figures significantly enrich the work. The discussion is very interesting and raises the issue of dispelling any doubts about the validity of this study. I have no comments on the manuscript - it is beautifully written and it is clear that the authors put a lot of work into conducting the research and preparing this manuscript. 

Many reviewers, which I also struggle with in my research, talk about small groups - but it should be noted that animal studies and obtaining consent from bioethics committees usually only allow for such numbers, so I fully understand the authors and they should be defended here if any reviewer raised the topic (because I know that usually people who do not conduct such research make such comments).

Author Response

Thank you very much for your review. We very much appreciate your comments and can unfortunately relate to the described situation.

Round 2

Reviewer 1 Report

Thanks for the response. 

It is disappointing that the authors are unable to specify the types of arrhythmias. Also, from the explanation it seems that the occurrence of arrhythmias was marked by (an) anesthesiologist(s) which could be subjective and therefore might not be reproducible. 

No comment

Author Response

Thank you for your review. As stated before, we regret that we cannot provide more information. We will definitely keep your suggestion in mind for further studies. Furthermore, we strongly disagree that the evaluation of an ecg by an anesthesiologist should be cause to determine the study to be not reproducible.